# Metabolic size scaling reflects growth performance effects on age-size relationships in mussels (*Mytilus galloprovincialis*)

**Irrintzi Ibarrola, Kristina Arranz, Pablo Markaide, Enrique Navarro***

Departamento de Genética, Antropología Física y Fisiología Animal, Facultad de Ciencia y Tecnología, Universidad del País Vasco/Euskal Herriko Unibertsitatea (UPV/EHU), Bilbao, Spain

* enrique.navarro@ehu.eus

**Data Availability Statement:** All relevant data are within the manuscript and its Supporting Information files.

## Abstract

Body-size scaling of metabolic rate in animals is typically allometric, with mass exponents that vary to reflect differences in the physiological status of organisms of both endogenous and environmental origin. Regarding the intraspecific analysis of this relationship in bivalve molluscs, one important source of metabolic variation comes from the large inter-individual differences in growth performance characteristic of this group. In the present study, we aimed to address the association of growth rate differences recorded among individual mussels (*Mytilus galloprovincialis*) with variable levels of the standard metabolic rate (SMR) resulting in growth-dependent shift in size scaling relationships. SMR was measured in mussels of different sizes and allometric functions fitting SMR vs. body-mass relationships were compared both inter- and intra-individually. The results revealed a metabolic component (the overhead of growth) attributable to the differential costs of maintenance of feeding and digestion structures between fast and slow growers; these costs were estimated to amount to a 3% increase in SMR per unit of increment in the weight specific growth rate. Scaling exponents computed for intraindividual SMR vs body-mass relationships had a common value b = 0.79 (~ ¾); however, when metabolic effects caused by differential growth were discounted, this value declined to 0.67 (= ⅔), characteristic of surface dependent processes. This last value of the scaling exponent was also recorded for the interindividual relationships of both standard and routine metabolic rates (SMR and RMR) after long-lasting maintenance of mussels under optimal uniform conditions in the laboratory. The above results were interpreted based on the metabolic level boundaries (MLB) hypothesis.

## Introduction

Metabolism constitutes an integrated system of the energy yielding and utilization processes involved in supporting life; and most activities of organisms are reflected in their metabolic rates. The relationship between metabolism and the size of individuals has been the subject of thorough analysis for decades as a particularly meaningful case among the basic organismal attributes stemming from structural constraints set on functional properties. Partly for historical reasons, most of the studies on this subject have focused on animals and they rely on

**Funding:** Funder1 MINECO (https://sede.mineco.
gob.es) Project FIGEBIV (AGL2013-49144-C3-1-R)
Awarded: E.N., I.I. and P.M. Funder2 UPV/EHU
(www.ehu.es) Project: GIU20_064 Awarded: I.I.
and K.A. The funders had no role in study design,
data collection and analysis, decision to publish, or
preparation of the manuscript.

**Competing interests:** The authors have declared
that no competing interest exist.

determinations of the standard metabolic level based on oxygen consumption recorded under resting post-absorptive conditions, which is assumed to account for the energy requirements of both tissue maintenance and basic processes of homeostatic regulation [1]. As such, this metabolic level is considered as a standard for comparative purposes in multiple studies involving both inter- and intra-specific analyses.

Body-size scaling of metabolism in animals is typically allometric, expressed by a power function in the form: $R = a W^b$, where R is the metabolic rate, W is the body mass, $a$ is the scaling coefficient (or proportionality constant), and $b$ is the scaling exponent (or slope of the log-log relationship). The vast majority of these scaling exponents reported in animals in both inter- and intra-specific comparisons is approximately close to ⅔ or ¾, but is significantly lower than 1, implying that weight-specific metabolic rate (i.e. per unit of body mass) decreases with the increasing body size that encompasses both ontogenetic development and the evolutionary processes underlying wide-range speciation. This metabolic restriction imposed by size increments is a fundamental trade-off in all biological processes and is one of the topics of the most persistent and intense debates in the subject. Earlier studies have considered $b$ to approach the value ¾ in a variety of animals [2–5]; subsequently, this trend was interpreted as the expression of universal properties of resource-transport networks underlying a "¾ power law" for metabolism that is applicable to virtually all organisms [6–10]. However, several studies have recently questioned the universality of the "¾ power law" based on extensive surveys of available data sets for metabolic size-scaling that showed $b$ values to exhibit substantial variation among taxa and physiological states [11–14]. A potential approach to address the variability in scaling exponents is to assume that different constraints act as boundaries in the various processes that make up the organism's metabolic rate [13, 15–17], which would hence have different scaling characteristics. A derivation of the Dynamic Energy Budget (DEB) theory [18] considers that the acquisition of both energy and resources is proportional to surface area, while energy demand is proportional to volume; thus, variable scaling exponents would result from the specific balances between both types of processes achieved under the different conditions experienced by organisms. This view is consistent with the observation that most metabolic scaling exponents are between ⅔ and 1 according to Glazier [13], who developed the metabolic level boundaries (MLB) hypothesis as an extension of this approach to explore the dependence of scaling exponents on the level of metabolic activity of organisms [13, 16]. The MLB hypothesis assumes the above differential mass scaling for supply and demand processes, and states that the relative weight of these boundary constraints depends on the metabolic level (L = metabolism per unit mass) to predict a complex (U-shaped) relationship between the two parameters $a$ and $b$ of the allometric equation for metabolism vs. body mass. The MLB approach has proven useful in interpreting variations in scaling exponents associated with a diversity of conditions that shift metabolic activity across successive levels of maintenance, resting, routine, and active states of metabolism [13, 14, 16, 19].

Intra-specific analysis of metabolic size scaling, recorded in specimens covering an ample size range within the population, forms the majority of cases in scaling exponent determination for two reasons: **a)** some of the explicative causes of metabolic scaling are better approached through intra-specific analysis where, in opposition to inter-specific comparisons, patters of body organization are conserved, thus "avoiding the phylogenetic effects that plague inter-specific analysis" [13] and **b)** precise knowledge of allometric scaling exponents is required for size standardisation in studies of metabolism, allowing to "subtract" the effects of body mass on metabolic rate in order to discern other influences such as growth rate. This especially applies to those studies based on comparing groups of individuals that may inherently be size-heterogeneous as a consequence of large inter-individual differences in growth performance.

This is the case for bivalve molluscs, where extremely high rates of endogenous variability in growth have been reported [20–26]. For instance, long-lasting maintenance of spats of different bivalve species under homogenous conditions in the laboratory resulted in the progressive size-differentiation of the individuals spanning over a range of 5- to 10-fold (eventually up to 30x) [25–31], thus revealing a strong growth component of possible genetic origin. Phenotypes that are segregated as fast and slow growing, using extreme groups of such size distribution, exhibit noticeable differences in physiological behaviour that accounts for differences in growth performance [25, 26, 28, 32].

As suggested by these cases, ontogenetic growth in bivalves would include two differentiated components: the sequential phases of ontogenetic development, roughly related to age, and the overlaid inter-individual variability in growth rate that is responsible for size-to-age difference between fast and slow growers. One important point is that both sources of size variation in the population may generate different size constraints on standard metabolism and thus contribute differently to scaling effects. Consequently, an accurate characterisation of intra-specific scaling relations would require an analytical approach in which mass exponents fitted to the full size range of the population might be compared with those reflecting pure ontogenetic effects as based on the intra-individual size variation with age. Despite obvious interest [13], this type of approach has been addressed on very few occasions [33–35].

In this study, we analysed the size scaling of respiratory metabolism in individual mussels—*Mytilus galloprovincialis*—of a population. This study was mainly based on standard metabolic rate (SMR) determinations and aimed to differentiate two sources of individual size variation (time vs. rate of growth) as regard their effects on metabolic size scaling. Mussels from a uniformly-sized sample were allowed to differentiate in size under uniform feeding and thermal conditions in the laboratory, during which, their SMR and individual growth rate were recorded. On the other hand, SMR was determined in a heterogeneous sample of mussels covering the full size range found in the population. Parameters of allometric equations relating metabolism to body mass were then compared for two sets of data: **a)** intraindividual allometries with size ranges given by body-mass increments achieved during the growth period and **b)** inter-individual (intra-specific) allometries with size ranges given by natural size dispersion within the population.

## Material and methods

### Experimental design

Over 400 mussels were collected from monolayer mussel beds grown in a rocky intertidal area of Antzorape (Ibarrangelua, Bizkaia, Spain) on February 2013. Mussels ranged from approximately 7–35 mm in shell length, representing the minimum and maximum sizes observed in the bed, respectively. Mussels were placed inside tanks containing seawater set in a recirculating system regulated at ambient temperature (15°C) and salinity (34 psu), and fed a ration of *Isochrysis galbana* in 2 mm$^3$ particulate volume per litre (equivalent to 1.5 mg POM L$^{-1}$) continuously for two weeks. Subsequently, two experimental groups of mussels were created and were maintained in separate tanks for approximately seven months (February–August) under the above-mentioned temperature and feeding conditions.

**Group 1.** One hundred juvenile mussels of uniform size 10 mm (9.189 ± 0.907) were arranged in individual numbered chambers for six months (from March 5th to August 12th). The Shell length (SL), live weight including shell mass (LW), and SMR of each individual were recorded on eight occasions (approximately every three weeks). These measurements allowed for the calculation of individual growth rates of selected mussels and establishment of allometric scaling of metabolic rate with body weight at two different levels:

a. Intra-individual allometry: The scaling of SMR with LW for each individual mussel was determined by fitting power functions to metabolism and body size data recorded on eight occasions (n = 8). As a result, a series of 100 functions (k = 100) was obtained.

b. Inter-individual allometry: Similarly, for each sampling occasion, power functions were fitted to the SMR vs. LW data for the full sample. As a result, a series of eight power functions relating the metabolic rate to body size (k = 8) was obtained for n = 100 mussels.

On the 7th sampling date, additional measurements of routine metabolic rate (RMR) and clearance rate (CR) were individually determined (n = 100), and allometric scaling of these two parameters with LW was established as described for SMR.

Subsequently, the gill surface area was recorded in 30 selected mussels covering an ample range of body sizes to establish the allometric relationship of this parameter with the LW.

**Group 2.** Fifty mussels covering the size range observed in the sampled population (7–35 mm SL) were used to analyse the intra-specific allometric relationship between SMR and LW. Allometric relationships were analysed on two occasions: a) immediately after the group was created (*Intra-specific 1*: 18th of February) and b) four months post their maintenance in the laboratory under constant water temperature and continuous food supply (*Intra-specific 2*: 17th of June).

## Determination of SL, LW, and growth rate of juvenile mussels

On each sampling date, mussels were removed from the seawater tanks and carefully dried with tissue paper. Individual LWs (g) were determined in a $10^{-4}$ g precision balance and SL (mm) was measured using digital callipers. These measurements allowed the computation of individual growth rates for different time intervals over a 6-month period. Growth rate in terms of LW was expressed as i) *total growth rate* ($GR_{LW}$: g day$^{-1}$): individual LW increment per day and ii) *size-specific growth rate* ($SGR_{LW}$: %): $GR_{LW}$ divided by the initial LW of each interval, expressed as a percentage.

## Experimental determination of physiological parameters

**Clearance rate (CR; L h$^{-1}$).** CR determinations were performed in a static system by recording the exponential decay in particle concentration over time [36]. Mussels were individually placed in glass flasks (volume varied from 0.5 to 1.0 L according to mussel size) containing aerated sea water and particles of *Isochrysis galbana* ($\approx$40 particles µl$^{-1}$). Particle concentration was measured every 10 minutes for 1–2 hours with a Coulter Counter Z1 analyser. A control chamber without mussels was used for correction of the sedimentation rate of the particles.

**Metabolic rates.** Both SMR and RMR were estimated as the rate of oxygen consumption ($VO_2$: $mlO_2 h^{-1}$). RMR was measured after the fed mussels were directly transferred from the feeding tanks to respirometers. For SMR determinations (the majority of $VO_2$ recordings in this work), mussels were transferred to tanks filled with aerated filtered (1 µm) sea water and starved for three days before determination of oxygen consumption. Individual mussels were confined in chambers (the size of the chamber varied between 30 and 150 ml according to the size of the mussel) sealed with luminescent dissolved oxygen probes connected to oximeters (HATCH HQ 40d); oxygen consumption rates were estimated from the decrease in oxygen concentration over time (4–8 h). Control chambers without mussels were used to check the stability of oxygen concentration during the measurement period. In cases where both metabolic levels were determined, the metabolic scope for feeding and growth (MSFG; [26]) was calculated as the difference between RMR and SMR.

## Determination of gill-surface area (GA: mm$^2$)

At the end of the experiments, 30 mussels from *Group 1*, covering the entire size range of the group, were dissected by cutting their adductor muscles to expose the gills. One outer demi-branch of each individual was photographed with a digital camera placed next to a piece of graph paper to set the scale, and the area of the demibranch was determined using ImageJ software (National Institutes of Health; Bethesda, MD, USA). The areas estimated in this manner were doubled to account for each side of the demibranch and multiplied by the number of demibranchs (4) to estimate the gill surface area (GA; mm$^2$).

## Statistical procedures

One-way analysis of variance (ANOVA) was performed to test significance of differences in growth rate recorded for the different time intervals along the period of study. Post-hoc test (Tukey) was then applied to identify growth phases based on significant mean differences. Inter-individual and intra-individual allometric relationships between the SMR and LW were expressed according to the expression SMR = LW$^b$. The proportionality constant *a* and mass-exponents *b* were obtained by fitting logarithmically transformed individual SMR and LW data with regression equations. Mass-exponents (slopes) and proportionality constants (elevations) obtained in both the intra- and inter-individual treatments were compared using analysis of covariance (ANCOVA). The null hypothesis (H$_0$) with equal slopes ($b_1 = b_2 = b_3. . .. = b_k$) were tested using the F-statistic value. If H$_0$ was rejected, Tukey's multiple comparison test was performed to determine the significant differences between each pair of slopes. If H$_0$ is accepted, a common slope $b_c$ is computed and the null hypothesis (H$_0$) of equal elevations ($a_1 = a_2 = a_3. . .. = a_k$) were subsequently tested using the F-statistic value. If H$_0$ was accepted, then a common elevation $a_c$ and common regression were computed. If H$_0$ was rejected, Tukey's multiple comparison test was performed to determine the significance between each pair of elevations. All statistical analyses were performed using a custom R script based on the procedures described by Zar [37].

Complementarily, SMR data used in both the intra- and inter-individual analyses of *Group 1* mussels were plotted against multiple factors, using LW, growth rate (SGR$_{LW}$), and the interaction term as potential predictors, and a simultaneous function was fitted by multiple regression procedures using SPSS (IBM SPSS Statistics V. 25).

## Results

### Group 1: Mussels of initial uniform size

**Growth and size distribution.**   The time-course of size change of 100 mussels during 150 days of maintenance in the laboratory is shown in Fig 1A and 1B for both LW and SL. Different size trajectories were illustrated by the progressive divergence of these lines, accounting for great differences in growth rates among individuals. An almost continuous decline in values for condition index (CI: mg LW /mm SL$^3$), from >0.1 to <0.1 (Fig 1C), indicated that LW increments are majorly driven by shell growth.

Underlying inter-individual variability, sequential phases of fast/slow growth were recorded for growth rates computed for LW (Fig 2A), with average values of 7 and 12 mg day$^{-1}$ for the early and late phases of fast growth, respectively, and a minimum average of 3 mg day$^{-1}$ in the slow growth phases. Change in growth trend was particularly intense from day 84 to 119, with growth rates increasing from 2.5 to 17 mg day$^{-1}$. When growth is expressed as a weight-specific rate (i.e., as a percentage of daily increment per unit weight or SGR$_{LW}$) (Fig 2B), all individual data points can be fitted to a negative exponential function with time (curve in Fig 2B),

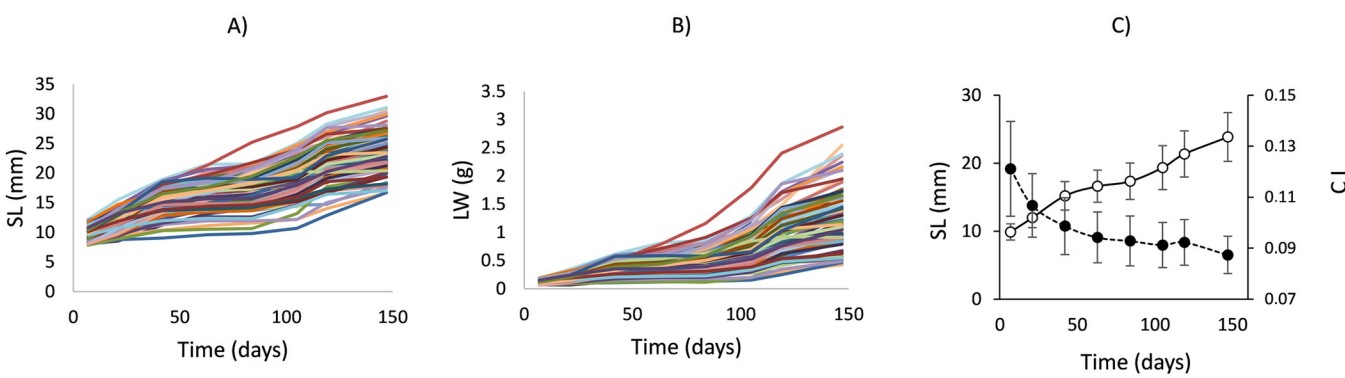

**Fig 1.** Growth trends of mussels (*Mytilus galloprovincialis*) from *Group 1* during the maintenance period, given in terms of: A) Shell length (SL: mm) and B) Live weight (LW: g); C) mean values of shell length (hollow symbols) and condition index (CI: LW / SL³) (full symbols). Bars represent ± SE.

representing the natural decline in the specific growth rate with size increment. Phases of fast/slow growth can be identified by points departing from the overall trend described by the curve, especially those corresponding to days 42 (4%), 84 (0.58%), and 119 (2.7%). After 119 days till the end of the experiment, there was a decrease in growth rate (both in absolute and relative terms) that might represent the onset of a subsequent period of slow growth, in agreement with the observed ~ 6-week (40 days) sequence of differential phases of growth.

The frequency distribution of $SGR_{LW}$ calculated for the entire growth period (from day 7 to 147) is plotted in Fig 3 on a logarithmic scale for normalisation purposes, removing skewness. Specific growth rates ranged from approximately 1–20% of initial LW increment per day. Fourteen individuals (approximately 15% of the group) displayed the lowest $GRW_{SPC}$ of 1.0–3.4% LW day$^{-1}$ and were considered a group of slow growers (S), whereas the 11 other mussels achieved the highest $GRW_{SPC}$ values, from 11.5 to 21.0% LW day$^{-1}$ and were considered a

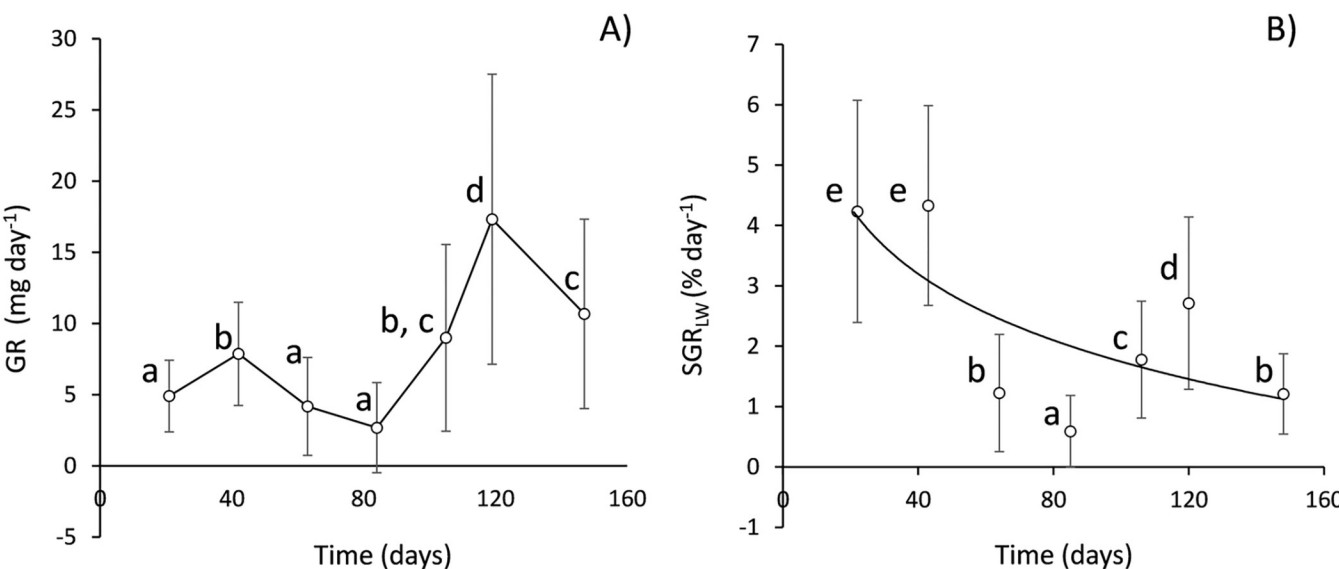

**Fig 2.** Mean rates of daily growth along the period of study, given in both A) absolute terms (GR: mg LW day$^{-1}$) and B) relative weight specific terms (SGR: % day$^{-1}$). Error bars represent ±SE. For reference, an exponential function was fitted to all individual data for SGR vs time (line in B): SGR = 3.621 (±2.189) * e$^{(-0.008 ± 0.006)* \ Time}$. Results of post-hoc test following ANOVA: Different letters denote significant mean differences.

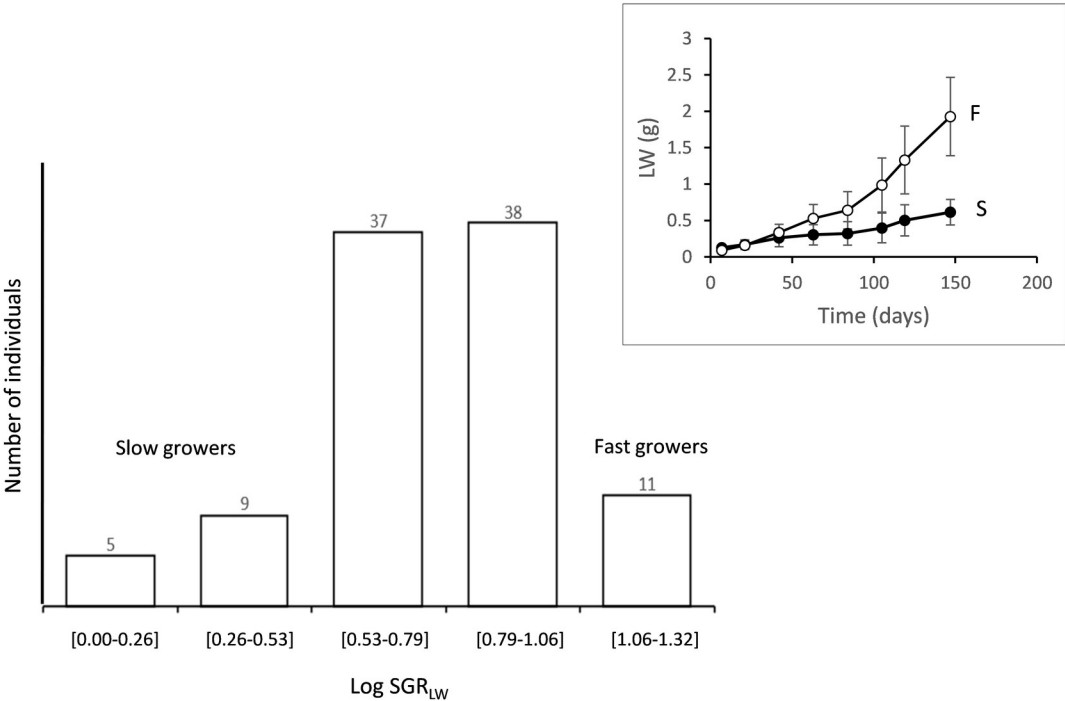

**Fig 3. Frequency distribution vs growth intervals.** For normality, SGR values were logarithmically transformed. Two groups of slow (S) and fast (F) growers were created with the extremes of the distribution. Insert: Growth trends of segregated groups of fast (F) and slow (S) growing mussels. Data are mean live weight (LW) values ± SE.

group of fast growers (F). Thus, the same range interval (1.8x) was accomplished in the selection of either F or S groups, and the computed growth rate difference between the two groups was 8.5x on average. A comparison of growth trends for both groups (Fig 3) indicated differences in the intensity of changes during the transition between growth phases, with S individuals showing almost nil growth during the slow growth phase.

**Intra-individual allometries of SMR.** Allometric equations relating SMR to LW for 100 individuals in *Group 1* were fitted using linear regression after log transformation of both variables (Fig 4). LW range was achieved through the growth of the individuals and varied between ~ 400 mg in the S group (LW-range of ~5x) and ~ 1800 mg in the F group (LW-range of ~20x). Only 5 out of the 100 individual regressions were found not significant (P > 0.05) and were excluded from further analysis.

The minimum and maximum values of the scaling exponents of the SMR were 0.570 ± 0.080 and 1.171 ± 0.127, respectively. Possible significant differences among the slopes and elevations of the 95 intra-individual allometries were analysed using ANCOVA. No significant differences in either the slopes or elevations were found (Table 1), indicating that all individuals share a common mass-exponent and proportionality constant.

Results of ANCOVA analysis for testing significant differences between slopes and elevations of the 95 intra-individual allometric relationship (on log-log scale) between standard metabolic rate ($VO_2$: ml $O_2$ h$^{-1}$) and body weight (LW: g). Size ranges varied between a minimum 5x to a maximum 30x. For about half these relationships, ranges varied between 10 and 20x.

Therefore, a common equation accounting for intra-individual scaling of SMR to body weight was calculated as follows:

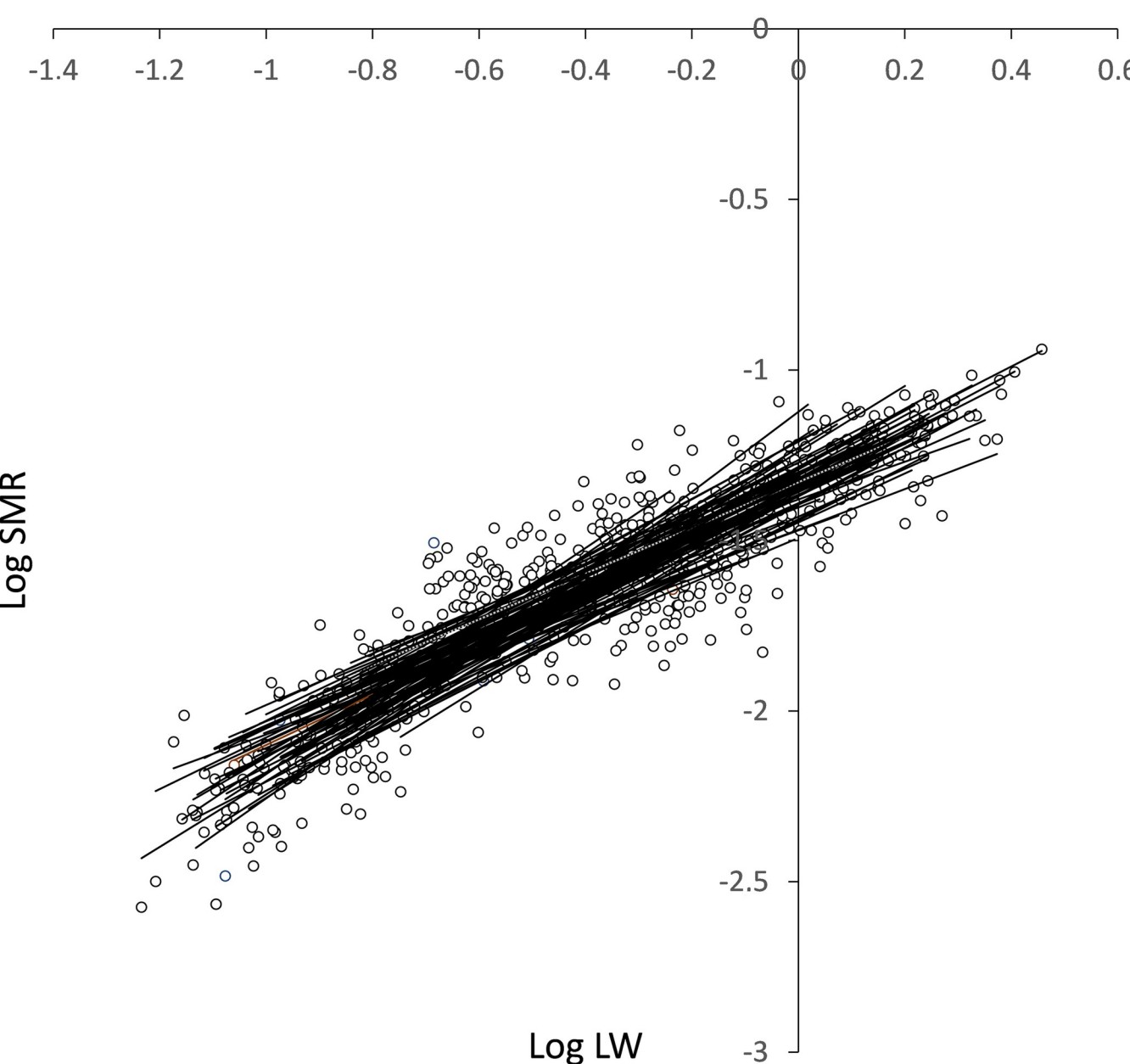

**Fig 4. Regression lines fitted to log-log transformed data of SMR (ml O$_2$ h$^{-1}$) vs LW (g) in 100 individuals.** Size ranges correspond to the weight increment experienced by each individual mussel. Only 5 of these regressions were not significant (P > 0.05).

**Table 1. Analysis of covariance (ANCOVA) of intra-individual SMR vs LW relationships.**

| Term of comparison | Slopes | Elevations |
|---|---|---|
| Hypothesis | H$_0$ = b$_1$ = b$_2$ = . . . = b$_{95}$ | H$_0$ = a$_1$ = a$_2$ = . . . = a$_{95}$ |
| F tab | 1.278 (0.05, (1), 94, 579) | 1.275 (0.05, (1), 94, 673) |
| F | 0.752 | 1.257 |
| P | 0.956 | 0.063 |
| Conclusion | Do NOT reject H$_0$ | Do NOT reject H$_0$ |
| Common value | b$_c$ = 0.789 | a$_c$ = -1.331 |

- Log SMR = 0.789 (± 0.013) × Log W− 1.331 (± 0.007);

$$R^2 = 0.824; \ P < 0.0001 \tag{1}$$

The mean values (± SD) of the slopes $b$ and elevations $log\ a$ are plotted in Fig 5. A significant correlation ($R^2 = 0.61$; P < 0.001) was found between these two parameters (Fig 5A), indicating a positive effect of metabolic level on the steepness of the lines scaling SMR to body size. However, neither of the two parameters correlated with individual growth rates (weight-specific values) (Fig 5B and 5C), very likely due to heteroskedastic distribution of variances along the SGR axis, where larger variances in slow growers (low SGR) can be attributed to the narrower size-ranges over which regression analysis were deployed compared with fast growers (high SGR). This poses a methodological constraint on the attempt of exploring the effects of individual growth rates on rates of metabolism.

Thus, a further attempt was made by fitting a multiple regression model in which the SMR of each individual on a given sampling date was given as a simultaneous function of both the body weight and the weight-specific growth rate ($SGR_{LW}$) recorded on that date. Together with these independent variables, potential predictors in the model also included the combination of body weight and growth rate, although this interaction term did not significantly affect the SMR. The fitted equation (n = 665) with the two remaining variables is:

- Log SMR = 0.677 (± 0.016) × Log W + 0.014 (± 0.003) × SGRLW− 1.382 (± 0.009)

$$R^2 = 0.745; \ F: \ 921.3; \ P < 0.0001 \tag{2}$$

The weight-spec(3)ific growth rate exhibited by individuals exerted, along with body size, highly significant positive effects on SMR, and thus contributed to a more detailed description of factors affecting SMR. Particularly, this model accounts for the finding that, after the metabolic effects of growth were subsumed in the corresponding coefficient, body size scaling exponents were found to decrease from 0.79 (1) to 0.68 (2).

**Inter-individual allometries of standard metabolic rates.** Inter-individual allometric relationships between SMR and LW were fitted using the size distribution of 95 mussels in *Group 1* on the eight sampling dates during the growth period. The parameters of the fitted log-transformed equations are presented in Table 2, together with size ranges that varied between a minimum (3.3x) on day 7 and a maximum (18.1x) on day 147. ANCOVA results

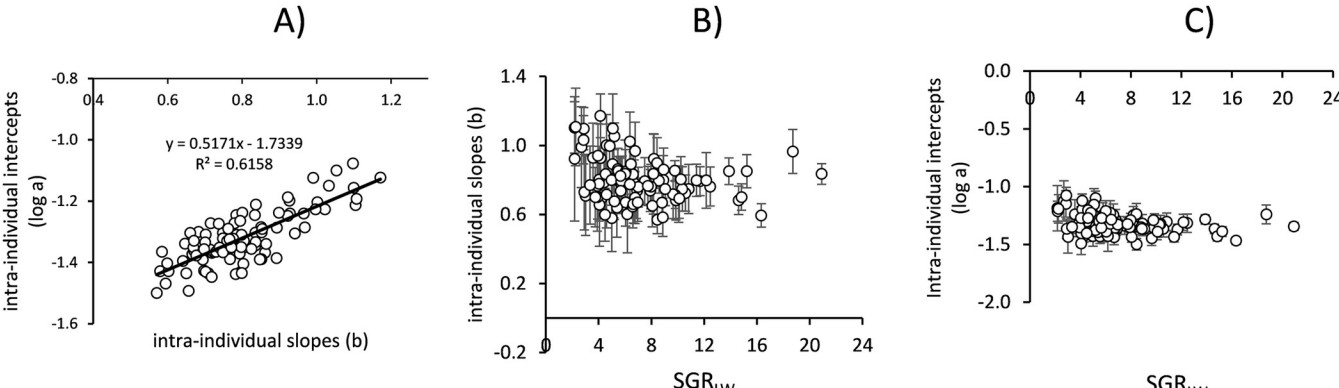

**Fig 5.** A) Relationship between proportionality constant (*log a*) and mass exponents (*b*) of individual allometric relationships of standard oxygen consumption and live weight. B) Mass-exponents *b* (±SD) and C) Proportionality constant, *log a* (±SD) of intra-individual allometric relationships of standard oxygen consumption and live weight plotted as a function of specific growth rate of live weight ($SGR_{LW}$).

**Table 2. Inter-individual allometric relationships in different growth phases.**

| Day | Range LW(g) | B | a | R² | P |
|---|---|---|---|---|---|
| **7** | **0.058–0.190** | **0.777 (±0.091)** | **-1.45 (±0.087)** | **0.444** | **<0.0001** |
| 21 | 0.062–0.342 | 1.050 (±0.066) | -1.04 (± 0.051) | 0.723 | <0.0001 |
| 42 | 0.094–0.614 | 0.733 (±0.075) | -1.30 (± 0.038) | 0.501 | <0.0001 |
| 63 | 0.093–0.810 | 0.653 (±0.067) | -1.37 (±0.029) | 0.504 | <0.0001 |
| 84 | 0.096–1.161 | 0.627 (±0.073) | -1.42 (± 0.032) | 0.486 | <0.0001 |
| 105 | 0.145–1.791 | 0.800 (±0.048) | -1.33 (± 0.014) | 0.747 | <0.0001 |
| 119 | 0.153–2.406 | 0.701 (±0.067) | -1.39 (± 0.015) | 0.536 | <0.0001 |
| 147 | 0.155–2.806 | 0.682 (±0.044) | -1.33 (± 0.009) | 0.715 | <0.0001 |

Regression equations for the inter-individual allometric relationship of standard oxygen consumption and body mass in the 8 sampling dates. Range: live weight (g) range of the mussels. b: mass-exponent a: proportionality constant (± SE). ANCOVA testing for significant differences between slopes (b): df = 720; F = 3.0488; P = 0.003; Tukey test to compare between pair of slopes indicates that slope in sampling date 2 is significantly different to those in dates 4, 5, 7 and 8. Removing Eq 2, the ANCOVA indicates that the slopes in the remaining equations are not significantly different and the calculated common slope is 0.682.

revealed significant differences between slopes *b* of regression lines (Table 2), and the mean comparison (Tukey's test) indicated significantly higher mass exponent on day 21 (*b* = 1.05) than of most of the sampling points (4th, 5th, 7th, and 8th sampling dates). ANCOVA comparisons following the removal of this day 2 equation showed no significant differences between regression lines, with a common slope (*b* = 0.682).

As in the intra-individual approach, possible metabolic effects of growth differences were tested by multiple regression models, where SMR was plotted as a simultaneous function of LW and the weight-specific growth rate ($SGR_{LW}$). This analysis was separately performed in eight datasets obtained at different sampling dates along the growth period of mussels in *Group 1*. According to the fitted models, $SGR_{LW}$ exerted significant effects on only three out of the eight sampling dates. The resulting equations are as follows:

- Day 84 (5th sampling date):

$$\text{Log SMR} = 0.529 \, (\pm 0.085) \times \text{Log LW} + 0.062 \, (\pm 0.029) \times SGR_{LW} - 1.484 \, (\pm 0.031).$$

$$R^2 = 0.469; \ F = 32.64 \ P < 0.0001 \tag{3}$$

- Day 105 (6th sampling date):

$$\text{Log SMR} = 0.715 \, (\pm 0.053) \times \text{Log LW} + 0.030 \, (\pm 0.011) \times SGR_{LW} - 1.403 \, (\pm 0.028).$$

$$R^2 = 0.735; \ F = 127.26; \ P < 0.0001 \tag{4}$$

- Day 119 (7th Sampling date):

$$\text{Log SMR} = 0.686 \, (\pm 0.070) \times \text{Log LW} + 0.029 \, (\pm 0.009) \times SGR_{LW} - 1.461 \, (\pm 0.029).$$

$$R^2 = 0.529; \ F = 49.97; \ P < 0.0001 \tag{5}$$

Two features from this analysis are relevant: **i)** $SGR_{LW}$ appeared as a factor significantly affecting SMR on those dates encompassing a phase of clear growth improvement (Fig 2B), when the maximum size differentiation between F and S group individuals was attained (Fig 3) and **ii)** as seen before (**Intra-individual allometries of SMR**), the computation of a specific

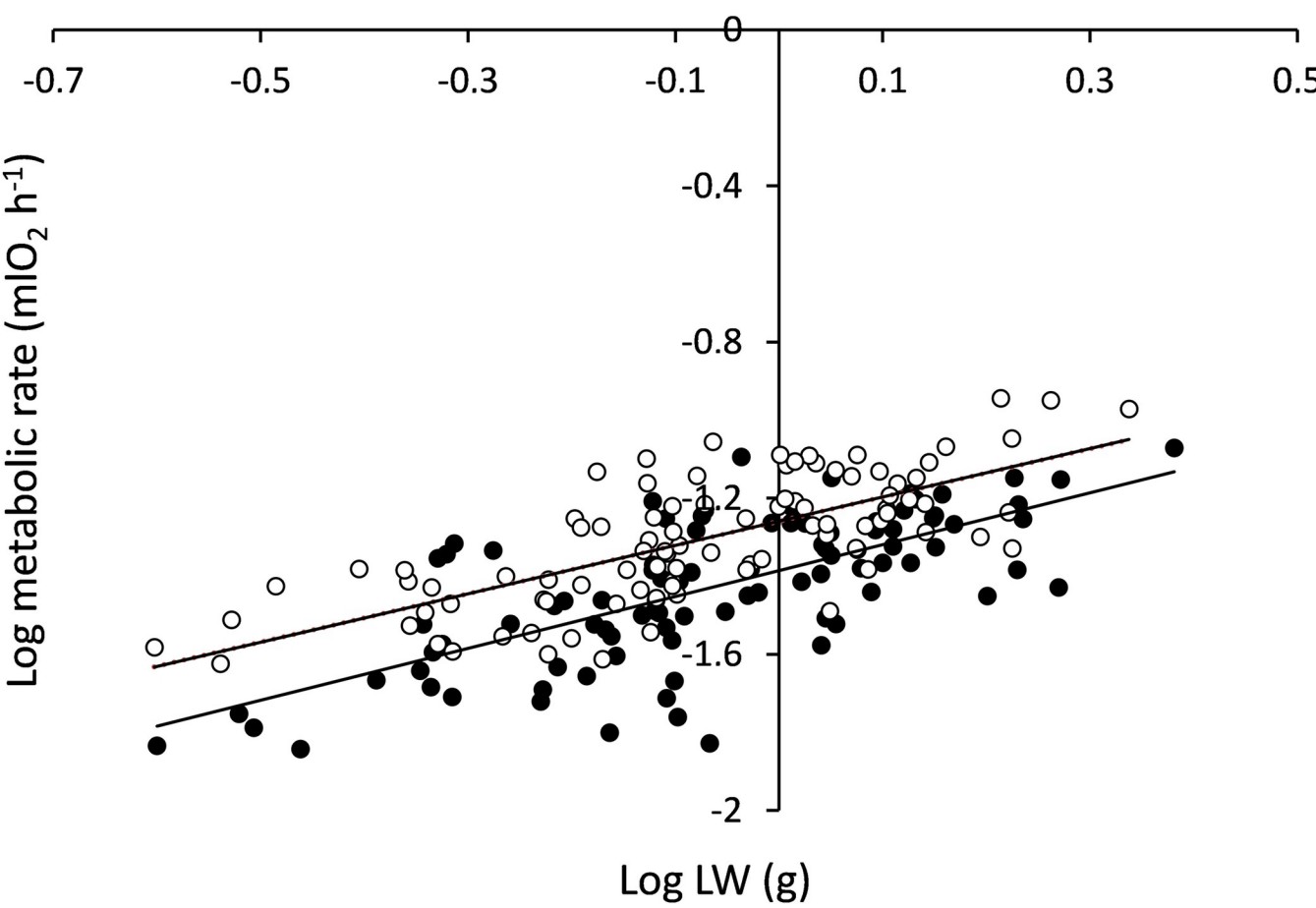

**Fig 6. Log-log regression lines for routine metabolic rate (RMR: Upper line; hollow symbols) and standard metabolic rate (SMR: Lower line; full symbols) vs live weight (LW).** Interindividual analysis performed on day 119 in mussels from *Group 1*.

coefficient accounting for the metabolic effects of growth results in a general reduction of the mass-exponents fitted with this multiple factor model compared with the corresponding b values reported in Table 2.

**Comparative inter-individual allometries for SMR and RMR, CR and gill area.** On the 7th sampling date (day 119), in addition to SMR, individual measurements included RMR and CR. Regression equations fitted to individual data for SMR and RMR vs. LW (Fig 6) for standard and routine rates are as follows:

- Log SMR = 0.701 ($\pm$ 0.074) Log LW– 1.388 ($\pm$ 0.015);

$$R^2 = 0.536; \ P < 0.001 \tag{6}$$

- Log RMR = 0.620 ($\pm$ 0.062) Log LW– 1.259 ($\pm$ 0.013);

$$R^2 = 0.520; \ P < 0.001 \tag{7}$$

The results of the performed ANCOVA, to test significant differences between slopes and elevations of the regression lines for both levels of metabolism, revealed no significant differences in slope ($t = 0.54$; $P > 0.5$), allowing computation of a common slope ($b_c = 0.647$), while significant differences were observed between elevations ($t = 7.18$; $P < 0.001$), with the line for RMR running parallel to and above that of SMR.

Similarly, the following allometric relationship was obtained for CR:

- Log CR = 0.652 ($\pm$ 0.100) Log LW – 0.536 ($\pm$ 0.021);

$$R^2 = 0.338, \; P < 0.0001 \qquad (8)$$

Therefore, the mass exponents for the CR (0.652) and metabolic rates (0.647) were not significantly different.

The allometric relationship of the gill surface area ($mm^2$) to LW (mg) based on 30 individuals covering the size range found at the end of the growth period was as follows:

- Log GA = 0.689 ($\pm$ 0.067) $\times$ Log W + 0.605 ($\pm$ 0.205);

$$R^2 = 0.789, \; P < 0.001 \qquad (9)$$

As expected, the scaling exponent (0.689) did not significantly differ from ⅔ (the surface/volume relationship) and was similar to that of physiological rates of feeding and metabolism.

## Group 2: Intra-specific allometries

The intra-specific size scaling of SMR was analysed in a field sample of 50 individuals covering the entire size range observed in the population (Group 2) on two different occasions: **i)** two weeks after collection (*intra-specific1*; LW range of 0.054–4.315 g) and **ii)** after four months of maintenance in the laboratory (*intra-specific2*; LW range: 0.068–8.201 g). The corresponding data points for SMR vs. LW are plotted in Fig 7 and fitted with power functions of the form SMR = LW$^b$. The log-log transformed equations are as follows:

- *Intra-specific1*: Log SMR = 0.813 ($\pm$ 0.030) Log LW – 1.311 ($\pm$ 0.016);

$$R^2 = 0.943; \; P < 0.0001. \qquad (10)$$

- *Intra-specific2*: Log SMR = 0.709 ($\pm$ 0.028) Log LW – 1.350 ($\pm$ 0.016);

$$R^2 = 0.949; \; P < 0.0001. \qquad (11)$$

The ANCOVA testing for significant differences in slopes between *Intra-specific1* and *Intra-specific2* revealed mass-exponents that were significantly different ($t = 3.965$; $t_{0.05} = 1.9867$; $P = 0.0002$).

## Discussion

### Inter-individual growth rate differences

The broad differences in growth rates recorded between mussel juveniles reared under identical laboratory conditions indicate a strong endogenous component in the growth performance of bivalves. The weight-specific growth rate in mussels from *Group 1* followed a normal distribution (Fig 3), with a maximum of 10-fold difference between the extremes of the distribution.

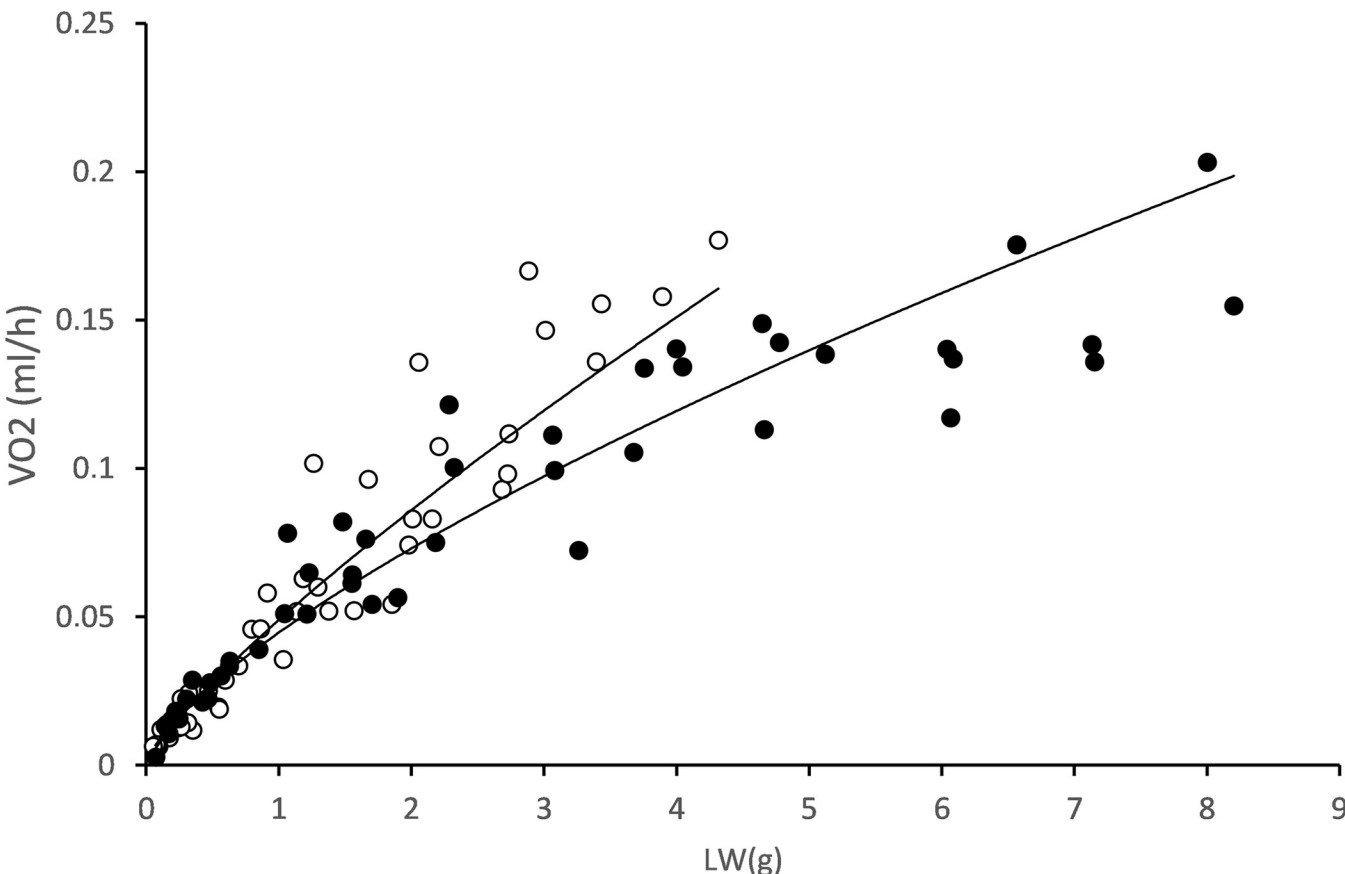

**Fig 7. Regression lines for the allometric relationships of the form SMR = a\*LW^b for mussels from *Group 2* after 2 wk.** (*Intraspecific 1*: upper line; hollow symbols) and 4 mo. (*Intra-specific2*: lower line; full symbols) maintenance in laboratory conditions.

We considered the size differentiation originating from this extreme growth variability as an opportunity to analyse the basis of growth allometry by comparing the size scaling of some physiological rates that are relevant to the quantification of energy balances. Thus, on the 7th sampling date, when mussels were size-differentiated after 5 months in the laboratory, CRs, RMRs, and SMRs were recorded in all the 95 individuals and plotted against the LWs for comparison of allometric exponents. Previous studies based on the simultaneous measurements of clearance and metabolic rates along the size range of bivalve populations have reported lower mass-exponents for CR than for metabolic rate [38–45], indicating that with increasing body weight, the mass-specific filtering activity decreases at a higher rate than the mass-specific metabolic rate; therefore, the energy balance tends to decrease with an increase in size, which provides a physiological interpretation for the age-related decline in growth efficiency. On the other hand, this kind of differential behaviour of CR vs. metabolic rate is interpreted on the assumption that SMR scales faster with body size than does RMR, which results in a smaller fraction of metabolic energy, the metabolic scope for feeding and growth (MSFG = RMR–SMR), which is available for these functions as the size increases. Our study, restricted to an early phase of the life-span, found no such differences in size scaling between standard and routine rates (Fig 7), which is consistent with the identity of mass exponents for feeding (CR) and metabolic activities ($b = 0.65$) and confirms previous reports concerning the scaling behaviour of metabolism in juveniles of this species [46]. These size relationships are the same

as those found in the scaling of gill area to body mass ($b$ = 0.68), revealing the underlying sur-face-area limits set on both the physiological processes of feeding and respiration. The lack of difference in the exponents for energy gain and loss implied that the scope for growth (SFG) remained independent of body size, accounting for the observed age-related decline in the weight-specific growth rate (per unit of body mass).

### Intra-individual allometry of SMR

According to the results of ANCOVA comparing allometric functions fitted to SMR vs. LW data during the growth period for 95 individuals in *Group 1*, all individuals shared a common allometric scaling exponent (0.789) which would indicate that, on average, the metabolic restrictions imposed by size evolution operate in the same way in all individuals of the group. However, large differences have been recorded among individuals of the group as regards to the age contribution to weight increment, which might result in ontogenetic effects on size scaling (aimed at the intra-individual approach) that appear biased by the effects associated with intrinsic interindividual variability in the weight-specific growth rate. Thus, a closer examination of the intra-individual distribution of allometric parameters (the mass exponent *b* and elevation *a*) regarding their dependence on weight-specific growth seems appropriate for disentangling possible mixed effects. Both allometric parameters fitted to intra-individual datasets appeared correlated ($R^2$ = 0.615; Fig 5A), suggesting a strong positive association between the level of metabolic activity of the individuals and their size dependency. On the other hand, the potential contribution of individual growth rate to intra-individual size scaling of metabolism was assessed by fitting all intra-individual SMRs data points (n = 665) with a multiple regression model of body mass and specific growth rate, as well as the interaction of both, as predictive variables. The resulting model (Eq 2) included a significant term for growth rate, accounting for a metabolic effect that can be calculated as an ~ 3% increment in SMR per unit increment in $SGR_{LW}$ (i.e. 1% of LW per day). Moreover, despite the lack of a significant interaction term, a comparison of the body-mass exponents in Eqs 1 and 2 indicates that when the effect of differential growth between individuals is discounted, the size-scaling b value for SMR drops from 0.79 to the 0.68, which is characteristic of surface-to-volume relationships. According to previous considerations (see **Introduction**), the latter could be considered as the allometric parameter governing the scaling of maintenance metabolism along an interval of body size increments driven only by an age-based sequence of ontogenetic development. The same results were obtained through similar multiple regression equations fitted to SMR vs. LW data from the inter-individual analysis with regard to both the existence of a specific posi-tive parameter accounting for the metabolic effects of growth rate and the associated decline in the mass exponent (*b* values in Table 2 are compared with the corresponding values in Eqs 3, 4 and 5).

Concerning the origin of a specific component of the SMR associated with variable growth performance of mussels, it should be noted that such metabolic fractions would not corre-spond to the "costs of growth" *sensu* Parry [47, 48], supposedly abolished along the starvation conditions prior to SMR measurements, unless in some indirect sense; i.e., as a reflection of the differential cost of maintenance of enlarged organs (gills and digestive glands) that likely support the greater levels of energy acquisition that underlie growth improvement (the over-head of growth [18]). In support of this statement, constitutive fast growers (F group individu-als in this study) have been reported to exhibit greater gill areas per unit body mass compared to slow growers (S group individuals in our study) in this species of mussel [26, 31] as well as in clams (*Ruditapes philippinarum*) [25]. Similarly, significant increases in SMR have been reported for mussels and clams conditioned to high food rations than in those fed low food

rations [28, 49], where improved growth observed in the former would partly rely on enlarged and more active digestive structures such as the digestive gland [50, 51].

To discuss the implications of the occurrence of this component with regard to the size scaling of SMR, we call on some concepts taken from the MLB hypothesis [16, 52–54]. According to this hypothesis, scaling exponents would reflect a shifting equilibrium—as a function of metabolic level—between tissue demands for metabolic activity, which are roughly proportional to body mass, and the constraints set by the resource supply and distribution systems obeying the ⅔ (surface-to-volume) rule. The prediction based on size dependency differences for the two components of SMR, (body tissue maintenance and the added costs of whole organism homeostatic regulation), is that in the event of an increased fraction of SMR being constituted by tissue maintenance costs, these demands would impose a shift in size scaling exponents departing from ⅔ toward strict proportionality or isometry ($b = 1$). This interpretation is consistent with our results from the intra-individual analysis, where **i)** a positive correlation ($r = 0.615$; $P < 0.0001$) was found relating $a$ (intercept) and $b$ (slope) parameters from intra-individual allometric equations and **ii)** SMR values deprived (cut off) from the specific component accounted for by differential growth were shown to scale with body weight according to a $b$ value 0.677 ($\sim$ ⅔), while $b = 0.79$ was the common weight exponent for full range SMR values.

## Inter-individual (intra-specific) allometry of SMR

Inter-individual allometric relationships of SMR were analysed in two ways in this study.

First, mussels in *Group 1* had a similar initial size and very likely belonged to the same cohort of the sampled population; thus, size range was mainly achieved through progressive size differentiation in the laboratory owing to endogenously determined inter-individual differences in growth performance. This was accounted for by the series of length/weight vs. time growth curves (Fig 1) and resulted in increasing size ranges over time (Table 2). The behaviour of F and S groups (Fig 3) clearly illustrated the basis of this size differentiation. The average slope for the allometric equations relating SMR to LW in this inter-individual analysis (Table 2) was 0.75; however, the parallelism test applied to these equations (ANCOVA) indicated significant differences in their slopes, with values for day 21 ($b = 1.05$) differing from those on days 63, 84, 119, and 147 having a common b = 0.66 (= ⅔). Given the occurrence of different growth phases during the maintenance period in the laboratory, it is suggested that these slope changes might reflect the specific metabolic effects of growth differences. Recalculation of the size exponents using multiple regression models, including a term for the specific growth rate, provided further confirmation of such effects; in the three instances in which significant growth effects were recorded (days 84, 105, and 119), models yielded $b$ values that were clearly reduced with respect to the original values. Thus, confirming the main results from the intra-individual analysis, the component of size differentiation due to growth variability is responsible for the increment in the scaling exponents of SMR. On the other hand, the value of the growth coefficient computed for day 84 (0.062) indicated a more intense effect of growth on SMR corresponding to a phase when growth differences attained a maximum. The magnitude of this differential effect can be evaluated by comparing the metabolic behaviour in F and S groups: from the multiple regression equation it was predicted that, in a standard 0.2 g LW-mussel, the computed increments of $GRW_{SPC}$ from 3% to 10% (mean specific growth rates for S and F mussels, respectively) would promote an increase in oxygen consumption from 21 to 58 $\mu$l h$^{-1}$. These are the maximal differences observed during the critical phase of slowing growth, where almost virtual cessation of growth in the S group mussels was associated with a strong reduction in SMR that did not occur in the F group mussels. According to

the values of growth coefficients for metabolism in other temporal phases, such F vs. S differences tended to lessen as growth activity was resumed. A similar behaviour of metabolic scaling along phases of ontogenetic growth have been reported for terrestrial gastropods, with slope values declining between juvenile (fast growth) and adult (slow growth) stages of development [55, 56].

Second, mussels in *Group 2* belonged to different cohorts of the population; therefore, the size variation in the allometric relationships (Fig 7) was mainly caused by age differences between individuals. In *Group 2*, the scaling exponent decreased significantly from 0.813 with mussels recently transferred to the laboratory from the natural intertidal environment (*intraspecific1*) to 0.709 with mussels acclimated for four months in the laboratory (*intraspecific2*). Such a shift in the scaling exponent might reflect the fact that inter-individual differences in growth conditions due to environmental factors (very likely associated with the trophic and thermal heterogeneity characteristic of the intertidal habitat) were attenuated after the 4-month maintenance of individuals under the optimal and stable conditions in the laboratory, thus reducing the original growth variability within this group, along with the already discussed effect of this variability on the size exponent. This interpretation is consistent with the hypothesis postulated by Glazier and collaborators [13, 54, 57, 58] that ecological factors such as the presence of predators or changes in available diet or thermal environment affect the scaling exponent of intra-specific allometry of SMR.

This scaling exponent of *Group 2* (differentially aged mussels) after acclimation to laboratory conditions ($b = 0.709$) differed significantly from the common exponent of lines fitted to intra-individual SMR values ($b = 0.79$) or the average value of slopes recorded in the inter-individual analysis of *Group 1* mussels in the initial period (2 months) of growth under laboratory conditions (Table 2), when endogenous inter-individual differences in growth rate were considered to greatly contribute to the size variation of this single cohort of mussels. However, irrespective of some irregularities represented by growth phases, these differential growth effects tended to decline with the age of mussels, as indicated by the general behaviour of weight-specific growth rates during the maintenance period (Fig 2B), with a corresponding decline in the slope from day 63 onwards to values (average $b = 0.68$) that were not significantly different from those for *Group 2* mussels ($b = 0.709$) or other reported values ($b = 0.71$) for a similar size range of the species [46].

## Concluding remarks

The scheme shown in Fig 8 was elaborated to help summarise some concluding points concerning size-scaling dependence on the variable levels of standard metabolism achieved by mussels exhibiting differences in growth conditions.

These conclusions have been mainly inferred from the results of the present analysis of intra-individual SMR vs. body weight relationships in 95 juvenile individuals, although they were also supported by different approaches to the intra-specific (inter-individual) relationships reported herein. For argumentative convenience, the full size-range in this scheme has been plotted as the addition of two components: the first component is strictly dependent on the time of growth (age) and the second component is associated with differential growth accounting for size differences among individuals of the same age. Curves for SMR vs. body size were then plotted for the two categories of F and S group individuals, where the level of demand for tissue maintenance constitutes the specific difference regarding resting metabolism, according to reasons that have already been discussed. The assumption is that for the S group individuals, metabolic demands for the homeostatic regulation of body functions (e.g., surface-related processes of resource supply and waste disposal) are a prevalent component of

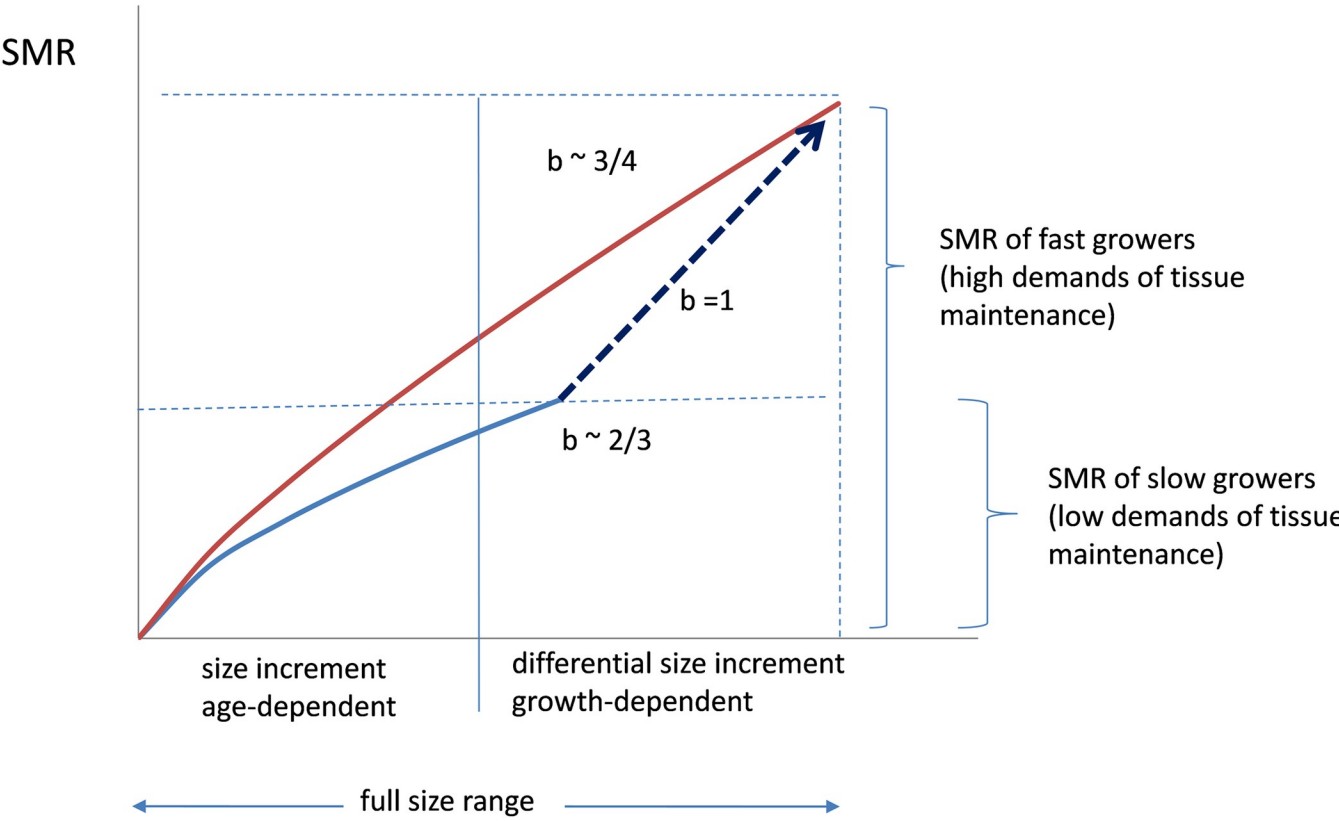

**Fig 8. Proposed scheme accounting for a shift in the scaling exponent for the standard metabolic rate vs body size relationship, between slow growers (low maintenance demands) and fast growers (high maintenance demands).** See text for details.

SMR over tissue maintenance, resulting in metabolic scaling of body size according to a surface/volume exponent ($b = \frac{2}{3}$). Conversely, in the F group individuals, SMR increment resulted from the higher demands for tissue maintenance. These demands are assumed to scale isometrically ($b = 1$) the size span achieved through differential growth, thus imposing a shift (as indicated by the arrow) in the weight exponent to values close to $b = \frac{3}{4}$. This shift was observed irrespective of the origin of differences in growth conditions, either endogenous (*Group 1* experiments) or exogenous (when maintenance in the laboratory of *Group 2* mussels tended to smooth the original variability in growth conditions of freshly collected specimens), indicating that variable demands for tissue maintenance constitute a basic feature inherent to differences in growth performance.

Since the present interpretation was based on assumptions underlying the MLB hypothesis [16] it is necessary to discuss how the present analysis supports the finding of a positive dependence in the relationship between the mass exponent and the level of resting metabolism (as represented by the slope and elevation of allometric functions in the intra-individual analysis; Fig 5A); while the opposite tendency (see Fig 2 in [16]) constitutes the core of MLB hypothesis predictions for resting metabolism, supported by empirical observations performed on a wide range of animal species, both ectotherms and endotherms [13, 16, 52]. The consideration of some distinctive features of bivalve molluscs, partly shared by other marine invertebrates of indeterminate growth, might help explain the above disagreement based on differences in the nature of variations in metabolic resting levels between the present case and the more general case presented by Glazier [13, 16]. In the latter, minimum values are assumed to represent

constant tissue maintenance requirements ($b = 1$); the addition of regulatory demands (including expensive homeothermy, when applicable) satisfied through fluxes of metabolic resources, wastes, and (or) heat, which are surface-area-limited ($b = ⅔$), to this minimum value would result in increased levels of the resting metabolism [16]. This is seen to differ in bivalve molluscs, whose growth heterosis has been rated among the highest in the animal kingdom [59], resulting in large inter-individual differences in growth rate, typically reaching up to 10-fold in the mussels of our study. Here, variable levels of resting metabolism would stem from growth-related differences in tissue maintenance requirements superimposed on the rather weak regulatory demands that is characteristic of conforming organisms. Isometric size dependence of this metabolic component will thus prevail in faster growers, accounting for higher mass exponents when SMR scales the size ranges that are mostly dependent on growth rate differences among individuals, as in the case of intra-individual analysis of a single cohort of juveniles subjected to progressive size differentiation in the laboratory. The progressive decline of this differential factor in SMR with the aging of mussels would result in mass exponents approaching the ⅔ value characteristic of metabolic demands driven by surface-to-volume-dependent processes. This also applies to additional demands set for activity, as in the metabolic scope for feeding and growth (MSFG) and is consistent with the $b$ value of 0.65 recorded for the RMR.

## Supporting information

**S1 Data.**
(XLSX)

## Acknowledgments

Authors are indebted to Douglas S. Glazier and one anonymous reviewer for valuable comments and suggestions that greatly contributed to improve the manuscript in the phase of revision.

## Author Contributions

**Conceptualization:** Irrintzi Ibarrola, Enrique Navarro.

**Data curation:** Kristina Arranz, Pablo Markaide.

**Formal analysis:** Irrintzi Ibarrola, Kristina Arranz, Pablo Markaide.

**Funding acquisition:** Irrintzi Ibarrola, Enrique Navarro.

**Investigation:** Irrintzi Ibarrola, Kristina Arranz, Enrique Navarro.

**Methodology:** Irrintzi Ibarrola, Kristina Arranz, Pablo Markaide.

**Project administration:** Enrique Navarro.

**Resources:** Kristina Arranz, Enrique Navarro.

**Software:** Pablo Markaide.

**Supervision:** Kristina Arranz, Pablo Markaide, Enrique Navarro.

**Writing – original draft:** Irrintzi Ibarrola.

**Writing – review & editing:** Enrique Navarro.

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
