## [Decision Letter · Decision Letter 0]

20 Jun 2022

PONE-D-22-11663Metabolic size scaling reflects growth performance effects on age-size relationships in mussels (Mytilus galloprovincialis).PLOS ONE

Dear Dr. Navarro,

Thank you for submitting your fine manuscript to PLOS ONE. After careful consideration, we feel that it has merit but does not yet fully meet PLOS ONE’s publication criteria as it currently stands. Therefore, we invite you to submit a revised version of the manuscript that addresses the points raised during the review process.

Both referees of the manuscript have made good suggestions for edits that will improve the manuscript. Both referees have also made suggestions regarding the statistical analyses and/or presentation of the statistical results. Please make these minor edits and changes before resubmitting your manuscript. Along these lines, I have two further suggested minor edits. 1) In figure 3, you use all capital letters. Please change these to normal case as in the rest of the figures.2) In Table 2, you use r2, but in the rest of the manuscript you use R2. Please correct the table to match the rest of the paper.

We look forward to receiving your revised manuscript.

Kind regards,

Erik V. Thuesen, Ph.D.

Academic Editor

PLOS ONE

Journal Requirements:

“This study was funded by the projects “Physiology and Genetics of Growth in Commercial Bivalves FIGEBIV” (AGL 2013-49144-C3-1-R) and GIU 20_064. P. Markaide was under a FIGEBIV research contract. K. Arranz was funded by a UPV-EHU predoctoral grant.”

“Funder1

MINECO (https://sede.mineco.gob.es)

Project FIGEBIV (AGL2013-49144-C3-1-R)

Awarded: E.N., I.I. and P.M.

Funder2

UPV/EHU (www.ehu.es)

Project: GIU20_064

Awarded: I.I. and K.A.

Reviewers' comments:

Reviewer's Responses to Questions

**Comments to the Author**

1. Is the manuscript technically sound, and do the data support the conclusions?

Reviewer #1: Yes

Reviewer #2: Yes

2. Has the statistical analysis been performed appropriately and rigorously? 

Reviewer #1: Yes

Reviewer #2: Yes

3. Have the authors made all data underlying the findings in their manuscript fully available?

Reviewer #1: Yes

Reviewer #2: Yes

4. Is the manuscript presented in an intelligible fashion and written in standard English?

Reviewer #1: Yes

Reviewer #2: Yes

5. Review Comments to the Author

Reviewer #1: General comments:

This thoughtful, multi-faceted study tested whether the mass-scaling of metabolic rate is related to growth rate in juvenile mussels. It is unique in making both intra- and inter-individual comparisons during different growth periods. As a result, the authors provide largely positive support for the idea that increased growth rates cause metabolic scaling exponents to increase significantly. They also show that under low growth conditions, the metabolic scaling exponent approximates 2/3, thus conforming to surface area to volume limits (as empirically demonstrated for gill surface area and food clearance rates). In general, the results are interpreted reasonably in terms of the metabolic-level boundaries hypothesis.

I have only two general comments/questions:

1) Did the authors compare the metabolic scaling of the slow vs. fast growers (as collective groups distinguished in their Figure 3)? This would seem to be a useful test of the idea that growth affects not only metabolic rate, but also its scaling with body mass.

2) In general, the authors appropriately cite much of the relevant literature, but I would suggest that they consider also citing the following studies, which specifically deal with relationships between growth rate and metabolic scaling in mollusks (albeit snails and not mussels):

Czarnołęski, M., Kozłowski, J., Dumiot, G., Bonnet, J. C., Mallard, J., & Dupont-Nivet, M. (2008). Scaling of metabolism in Helix aspersa snails: changes through ontogeny and response to selection for increased size. Journal of Experimental Biology, 211(3), 391-400.

Gaitán-Espitia, J. D., Bruning, A., Mondaca, F., & Nespolo, R. F. (2013). Intraspecific variation in the metabolic scaling exponent in ectotherms: testing the effect of latitudinal cline, ontogeny and transgenerational change in the land snail Cornu aspersum. Comparative Biochemistry and Physiology Part A: Molecular & Integrative Physiology, 165(2), 169-177.

Specific comments:

L 37: Change “metabolic level boundary” to “metabolic-level boundaries”.

L 105: To avoid confusion with genetically based evolution, I suggest changing “size evolution” to “ontogenetic growth”.

L 142: Please indicate that live weight (LW) includes shell mass, and not only living tissue.

L 228: To avoid confusion with genetically based evolution, I suggest changing “size evolution” to “growth”.

L 254: Change “other 11” to “11 other”.

L 405-407: How can a constant scope for growth (independent of body size) account for a decline in mass-specific growth rate with age?

L 517-518: The reference by McNab (2002) does not consider how ecological factors affect intraspecific metabolic scaling. McNab focuses on interspecific metabolic scaling, which he claims is fixed by “engineering” constraints.

L 552-555: Awkward wording. Please clarify.

L 558-560: Please clarify. The MLB hypothesis predicts that increasing activity should increase the metabolic scaling exponent.

L 560: Change “precedent” to “present”.

L 745: Change “2002” to “2020”.

Table 1: What are the size ranges for these intra-individual scaling relationships?

L 801, 806: Change “along” to “during”?

L 819: Change “were found no-significant” to “were not significant”.

Reviewer #2: The manuscript “Metabolic size scaling reflects growth performance effects on age-size relationships in mussels (Mytilus galloprovincialis)” investigates growth rates and mass-scaling patterns of metabolic rates in mussels. The study uses an interesting experimental design including both within- and among-individual analyses of growth rates and metabolic scaling. The authors find a gradient from slow to fast growth phenotypes across their experimental mussels. Although no significant differences are found in the mass-scaling of metabolism among individuals with slow or fast growth, the authors find that the overall mass-scaling exponent declines from ~3/4 to ~2/3 when growth rate is included in their mass-scaling models, matching the expected value under gill-surface dependent constraints.

Overall, I think study addresses a very interesting topic using an exciting experimental setting including both longitudinal and cross-sectional analyses of growth rates and mass-scaling of metabolic rates. The introduction adequately summarises the overall framework and main questions, and the discussion provides reasonable interpretations for the main results based upon previous research and theoretical frameworks such as the metabolic level boundary hypothesis. Hence, I only have minor suggestions that I think might facilitate the reading of the manuscript.

My main comment is that, although the metabolic level and scaling exponent appear unrelated to specific growth rates (SGR), the variance of both parameters declines towards higher SGR, with the scaling exponents becoming apparently more constrained around 2/3 as growth rate increases. I wonder if this reduction in variance indicates that higher energetic demands for growth ultimately constrains metabolic levels and scaling exponents to the values characteristic of surface dependent processes.

Lines 87-89: Awkward phrasing, probably a comma is required after “populations”.

Lines 92-96: I can’t follow the logical argument here: I understand that intra-specific analyses facilitate interpreting mass-scaling patterns of metabolic rates because body plan is conserved across individuals of the same species (point a), but why is “precise knowledge of allometric scaling exponents” required to study scaling exponents within populations (point b)?

Lines 155-157: Please include definitions of standard metabolic rate, resting metabolic rate and clearance rates.

Line 211: the variability in interindividual measurements (2-8) seems quite high considering that they are repeated measurements, was it due to mussel mortality? Also, the number of inter-individual samples in Group 1 was previously stated to be 100 individuals.

Lines 231-232: This sentence seems unclear, why did the authors envisage differential phases in growth rates? I think this result rather emerged a posteriori. In addition, determination of three phases in the continuous variation shown in Fig. 1c seems quite arbitrary. I would recommend including results of a post-hoc test illustrating among-group differences in growth rate.

Line 232: “were”.

Lines 233-235: So growth was primarily structural rather than due to reserve accumulation? Do the authors think that this pattern would differ later in life, e.g., before reproduction?

Lines 243-245: I agree that the observed deviations from the smooth exponential decay in growth rate can be used to interpret different phases of fast and slow growth; yet I would recommend performing a clustering post-hoc test to confirm that growth rate differed across three (or more?) different phases.

Line 268: Please provide regression coefficient as well.

Lines 346: Please provide P values for consistency with the rest of the Results section.

Line 422-423: both metabolic levels and slopes are unrelated to growth rate, but there is a striking reduction in variance towards higher SGR values. This suggests that metabolic levels and slopes become more constrained (slopes values around ~2/3) when growth rates are higher. Can the authors provide an interpretation of this pattern?

6. PLOS authors have the option to publish the peer review history of their article (what does this mean?). If published, this will include your full peer review and any attached files.

Reviewer #1: **Yes: **Douglas S. Glazier

Reviewer #2: No

---

## [Author Response · Author response to Decision Letter 0]

3 Aug 2022

Response to Review Comments to the Author

Reviewer #1: General comments:

This thoughtful, multi-faceted study tested whether the mass-scaling of metabolic rate is related to growth rate in juvenile mussels. It is unique in making both intra- and inter-individual comparisons during different growth periods. As a result, the authors provide largely positive support for the idea that increased growth rates cause metabolic scaling exponents to increase significantly. They also show that under low growth conditions, the metabolic scaling exponent approximates 2/3, thus conforming to surface area to volume limits (as empirically demonstrated for gill surface area and food clearance rates). In general, the results are interpreted reasonably in terms of the metabolic-level boundaries hypothesis.

Authors are extremely grateful to Reviewer for his favorable evaluation of the manuscript. 

I have only two general comments/questions:

1) Did the authors compare the metabolic scaling of the slow vs. fast growers (as collective groups distinguished in their Figure 3)? This would seem to be a useful test of the idea that growth affects not only metabolic rate, but also its scaling with body mass.

We did perform this comparison but with unclear results: Mean b value for intraindividual regression lines for the S (n = 14) was found higher than that for the F group (n = 11). However, we consider this might reflect a methodological bias derived from the extreme differences between both groups in the size range over which regressions are deployed (~5x in S compared with ~20x in F). Such effects of differences in size range associated to growth rate are evidenced in larger variances observed for both b and log a parameters, in slow compared to fast growing mussels (Figure 5) and precluded from finding a significant relationship of any of these parameters on the weight specific growth rate (SGR). It was this limitation the cause for trying multiple regression models in the attempt of fitting a specific parameter for SMR vs growth rate. We have included the consideration of these methodological issues (lines 285-306) 

2) In general, the authors appropriately cite much of the relevant literature, but I would suggest that they consider also citing the following studies, which specifically deal with relationships between growth rate and metabolic scaling in mollusks (albeit snails and not mussels):

Czarnołęski, M., Kozłowski, J., Dumiot, G., Bonnet, J. C., Mallard, J., & Dupont-Nivet, M. (2008). Scaling of metabolism in Helix aspersa snails: changes through ontogeny and response to selection for increased size. Journal of Experimental Biology, 211(3), 391-400.

Gaitán-Espitia, J. D., Bruning, A., Mondaca, F., & Nespolo, R. F. (2013). Intraspecific variation in the metabolic scaling exponent in ectotherms: testing the effect of latitudinal cline, ontogeny and transgenerational change in the land snail Cornu aspersum. Comparative Biochemistry and Physiology Part A: Molecular & Integrative Physiology, 165(2), 169-177.

We thank the referee for providing us with two very relevant references reporting effects of growth rate on metabolic scaling. Both references have been included (lines 544-547).

Specific comments:

L 37: Change “metabolic level boundary” to “metabolic-level boundaries”.

OK, done

L 105: To avoid confusion with genetically based evolution, I suggest changing “size evolution” to “ontogenetic growth”.

Ok, done

L 142: Please indicate that live weight (LW) includes shell mass, and not only living tissue.

Ok, done

L 228: To avoid confusion with genetically based evolution, I suggest changing “size evolution” to “growth”.

Ok, changed to “time-course of size change”

L 254: Change “other 11” to “11 other”.

Ok, done

L 405-407: How can a constant scope for growth (independent of body size) account for a decline in mass-specific growth rate with age?

The rationale is that the scope for growth represents absolute growth rates (e.g., mg day-1). Independency of SFG on body size would thus imply mass-specific growth rate to be reduced in older (= larger) individuals. 

L 517-518: The reference by McNab (2002) does not consider how ecological factors affect intraspecific metabolic scaling. McNab focuses on interspecific metabolic scaling, which he claims is fixed by “engineering” constraints.

Ok, this reference has been removed.

L 552-555: Awkward wording. Please clarify.

Ok, changed to: “Conversely, in the F group individuals increased SMR resulted from the higher demands for tissue maintenance. These demands were assumed to scale isometrically (b = 1) the size span achieved through differential growth, thus imposing a shift (as indicated by the arrow) in the weight exponent to values close to ¾. “

L 558-560: Please clarify. The MLB hypothesis predicts that increasing activity should increase the metabolic scaling exponent.

Yes, in the full range of metabolic activity. However, b would decrease with L for variable levels of resting metabolism (see Figure 2 in Glazier, 2010). I have included a reference to this figure in the text. 

L 560: Change “precedent” to “present”.

Ok, done

L 745: Change “2002” to “2020”.

Ok, done

Table 1: What are the size ranges for these intra-individual scaling relationships?

Size ranges varied between a minimum 5x and a maximum 30x.For about half these relationships, ranges varied between 10 and 20x. This information has been included in the Table 1 caption. 

L 801, 806: Change “along” to “during”?

Ok, done

L 819: Change “were found no-significant” to “were not significant”.

Ok, done.

Reviewer #2: The manuscript “Metabolic size scaling reflects growth performance effects on age-size relationships in mussels (Mytilus galloprovincialis)” investigates growth rates and mass-scaling patterns of metabolic rates in mussels. The study uses an interesting experimental design including both within- and among-individual analyses of growth rates and metabolic scaling. The authors find a gradient from slow to fast growth phenotypes across their experimental mussels. Although no significant differences are found in the mass-scaling of metabolism among individuals with slow or fast growth, the authors find that the overall mass-scaling exponent declines from ~3/4 to ~2/3 when growth rate is included in their mass-scaling models, matching the expected value under gill-surface dependent constraints.

Rather, the scaling exponent declines from 3/4 to 2/3 when metabolic effects of growth are “discounted” from the overall SMR vs body-mass relationship by fitting a multiple regression model. In fact, this would imply a positive effect of growth rate on mass scaling exponents. 

Overall, I think study addresses a very interesting topic using an exciting experimental setting including both longitudinal and cross-sectional analyses of growth rates and mass-scaling of metabolic rates. The introduction adequately summarises the overall framework and main questions, and the discussion provides reasonable interpretations for the main results based upon previous research and theoretical frameworks such as the metabolic level boundary hypothesis. Hence, I only have minor suggestions that I think might facilitate the reading of the manuscript.

Authors are indebted to Referee #2 for very appreciative comments on the manuscript

My main comment is that, although the metabolic level and scaling exponent appear unrelated to specific growth rates (SGR), the variance of both parameters declines towards higher SGR, with the scaling exponents becoming apparently more constrained around 2/3 as growth rate increases. I wonder if this reduction in variance indicates that higher energetic demands for growth ultimately constrains metabolic levels and scaling exponents to the values characteristic of surface dependent processes.

Higher variance for both parameters in slow growers is very likely related to reduced size ranges (~5x compared with >20x in fast growers) over which intraindividual allometric relationships were deployed (see my previous response to comments by Referee #1). Narrow size ranges would be expected to reduce the accuracy of fitting and, in fact, individual parameters could only be determined with a high variance in slow growers (see the amplitude or error bars in Figure 5). Hence, it appears that non homogeneous size-ranges, rather than metabolic demands per se, are responsible for this variance distribution along the SGR axis and explains why scaling exponents are constrained to a value characteristic of fast growing. However, this b value would be ¾ rather than 2/3 (see Figure 5).

As explained before, we consider this lack of size-range homogeneity as a methodological shortcoming that we tried to circumvent by using multiple regression models in the attempt to finding a specific metabolic term for growth rate. We have included the consideration of these methodological issues (lines 285-306). 

Lines 87-89: Awkward phrasing, probably a comma is required after “populations”.

Yes, a comma is lacking.

Lines 92-96: I can’t follow the logical argument here: I understand that intra-specific analyses facilitate interpreting mass-scaling patterns of metabolic rates because body plan is conserved across individuals of the same species (point a), but why is “precise knowledge of allometric scaling exponents” required to study scaling exponents within populations (point b)?

Point b reads as follows: “Precise knowledge of allometric scaling exponents is required for size standardization in studies of metabolism.” This means that it allows to “subtract” the effects of body mass on metabolic rate in order to discern other influences such as growth rate. I have completed the quoted paragraph in order to clarify this point. 

Lines 155-157: Please include definitions of standard metabolic rate, resting metabolic rate and clearance rates.

Standard metabolic rate is defined in the Introduction section. Resting and standard metabolic rates are generally used as synonymous. Clearance rate (volume of water cleared from particles per unit time) is a well established parameter among physiological measurements in suspension feeders. 

Line 211: the variability in interindividual measurements (2-8) seems quite high considering that they are repeated measurements, was it due to mussel mortality? Also, the number of inter-individual samples in Group 1 was previously stated to be 100 individuals.

There is a misunderstanding here caused by the very confuse way in which ANCOVA treatments were presented in the text. We thank the Referee for making this point. ANCOVA included comparisons for:

a) Intra-individual regression lines (k = 95; 5 out of the 100 regression lines were found not significant).

b) Interindividual regression lines for SMR vs RMR (k =2) on day 84.

c) Interindividual regression lines for SMR in 8 sampling dates along the growing period (k =8).

No mortality was recorded in mussels during present experiments and only metabolic data for 5 individuals from group 1 were not considered.

Description of statistical procedures concerning ANCOVA have been simplified in the manuscript to avoid misunderstanding. 

Lines 231-232: This sentence seems unclear, why did the authors envisage differential phases in growth rates? I think this result rather emerged a posteriori. In addition, determination of three phases in the continuous variation shown in Fig. 1c seems quite arbitrary. I would recommend including results of a post-hoc test illustrating among-group differences in growth rate.

“Denotate” is not the correct word: Differential phases of growth were in fact recorded. 

Line 232: “were”.

Ok, changed

Lines 233-235: So growth was primarily structural rather than due to reserve accumulation? Do the authors think that this pattern would differ later in life, e.g., before reproduction?

Positive. Mussels from Group 1 can be considered to be in the juvenile stage during the study. Condition index would be expected to increase with the onset of gametogenesis. 

Lines 243-245: I agree that the observed deviations from the smooth exponential decay in growth rate can be used to interpret different phases of fast and slow growth; yet I would recommend performing a clustering post-hoc test to confirm that growth rate differed across three (or more?) different phases.

Following your suggestion, post-hoc test following ANOVA were applied to identify growth phases based on significant mean differences. This treatment was performed on growth data and is reported in Figure 2. 

Line 268: Please provide regression coefficient as well.

Ok, done

Lines 346: Please provide P values for consistency with the rest of the Results section.

Ok, done

Line 422-423: both metabolic levels and slopes are unrelated to growth rate, but there is a striking reduction in variance towards higher SGR values. This suggests that metabolic levels and slopes become more constrained (slopes values around ~2/3) when growth rates are higher. Can the authors provide an interpretation of this pattern?

See my answer to your main comment.

---

## [Editor Report · Decision Letter 1]

9 Aug 2022

Metabolic size scaling reflects growth performance effects on age-size relationships in mussels (Mytilus galloprovincialis).

PONE-D-22-11663R1

Dear Dr. Navarro,

We’re pleased to inform you that your manuscript has been judged scientifically suitable for publication and will be formally accepted for publication once it meets all outstanding technical requirements.

Kind regards,

Erik V. Thuesen, Ph.D.

Academic Editor

PLOS ONE
---

## [Editor Report · Acceptance letter]

17 Aug 2022

PONE-D-22-11663R1 

Metabolic size scaling reflects growth performance effects on age-size relationships in mussels *(Mytilus galloprovincialis)*. 

Dear Dr. Navarro:

I'm pleased to inform you that your manuscript has been deemed suitable for publication in PLOS ONE. Congratulations! Your manuscript is now with our production department. 

Kind regards, 

on behalf of

Dr. Erik V. Thuesen 

Academic Editor

PLOS ONE